# Computational Investigation of 1, 3, 4 Oxadiazole Derivatives as Lead Inhibitors of VEGFR 2 in Comparison with EGFR: Density Functional Theory, Molecular Docking and Molecular Dynamics Simulation Studies

**DOI:** 10.3390/biom12111612

**Published:** 2022-11-01

**Authors:** Muhammad Sajjad Bilal, Syeda Abida Ejaz, Seema Zargar, Naveed Akhtar, Tanveer A. Wani, Naheed Riaz, Adullahi Tunde Aborode, Farhan Siddique, Nojood Altwaijry, Hamad M. Alkahtani, Haruna Isiyaku Umar

**Affiliations:** 1Department of Pharmaceutical Chemistry, Faculty of Pharmacy, The Islamia University of Bahawalpur, Bahawalpur 63100, Pakistan; 2Department of Biochemistry, College of Science, King Saud University, P.O. Box 22452, Riyadh 11451, Saudi Arabia; 3Department of Pharmaceutics, Faculty of Pharmacy, The Islamia University of Bahawalpur, Bahawalpur 63100, Pakistan; 4Department of Pharmaceutical Chemistry, College of Pharmacy, King Saud University, P.O. Box 2457, Riyadh 11451, Saudi Arabia; 5Department of Chemistry, Baghdad-ul-Jadeed Campus, The Islamia University of Bahawalpur, Bahawalpur 63100, Pakistan; 6Department of Chemistry, Mississippi State University, Starkville, MI 39759, USA; 7Laboratory of Organic Electronics, Department of Science and Technology, Linköping University, SE-60174 Norrkoping, Sweden; 8Department of Pharmaceutical Chemistry, Faculty of Pharmacy, Bahahuddian Zakariya University, Multan 60800, Pakistan; 9Molecular Biology and Bioinformatics Laboratory, Department of Biochemistry, Federal University of Technology, PMD 704, Akure 340106, Ondo State, Nigeria; 10Computer-Aided Therapeutic Discovery and Design (CATDD) Platform, Federal University of Technology, PMD 704, Akure 340106, Ondo State, Nigeria

**Keywords:** 1,3,4 oxadiazoles, epidermal growth factor receptor (EGFR), vascular endothelial growth factor receptor (VEGFR), density functional theory (DFT), molecular dynamics simulations

## Abstract

Vascular endothelial growth factor (VEGF) is an angiogenic factor involved in tumor growth and metastasis. Gremlin has been proposed as a novel therapeutic pathway for the treatment of renal inflammatory diseases, acting via VEGFR 2 receptor. To date, most FDA-approved tyrosine kinase (TK) inhibitors have been reported as dual inhibitors of EGFR and VEGFR 2. The aim of the present study was to find the potent and selective inhibitor of VEGFR 2 specifically for the treatment of renal cancer. Fourteen previously identified anti-inflammatory compounds i.e., 1, 3, 4 oxadiazoles derivatives by our own group were selected for their anti-cancer potential, targeting the tyrosine kinase (TK) domain of VEGFR2 and EGFR. A detailed virtual screening-based study was designed *viz* density functional theory (DFT) study to find the compounds’ stability and reactivity, molecular docking for estimating binding affinity, SeeSAR analysis and molecular dynamic simulations to confirm protein ligand complex stability and ADMET properties to find the pharmacokinetic profile of all compounds. The DFT results suggested that among all the derivatives, the **7g**, **7j**, and **7l** were chemically reactive and stable derivatives. The optimized structures obtained from the DFTs were further selected for molecular docking, and the results suggested that **7g**, **7j** and **7l** derivatives as the best inhibitors of VEGFR 2 with binding energy values −46.32, −48.89 and −45.01 kJ/mol. The Estimated inhibition constant (IC_50_) of hit compound **7j** (0.009 µM) and simulation studies of its complexes confirms its high potency and best inhibitor of VEGFR2. All the derivatives were also docked with EGFR, where they showed weak binding energies and poor interactions, important compound **7g**, **7j** and **7i** exhibited binding energy of −31.01, −33.23 and −34.19 kJ/mol respectively. Furthermore, the anticancer potential of the derivatives was confirmed by cell viability (MTT) assay using breast cancer and cervical cancer cell lines. At the end, the results of ADMET studies confirmed these derivatives as drug like candidates. Conclusively, the current study suggested substituted oxadiazoles as the potential anticancer compounds which exhibited more selectivity towards VEGFR2 in comparison to EGFR. Therefore, the identified lead molecules can be used for the synthesis of more potent derivatives of VEGFR2, along with extensive in vitro and in vivo experiments, that can be used to treat various cancers, especially renal cancers, and to prevent angiogenesis due to aberrant expression of VEGFR2.

## 1. Introduction

Cancer is one of the leading causes of mortality worldwide according to the World Health Organization (WHO) [1]. Among various cancer types, renal cancer is the seventh most common life-threatening disorder, causing 140,000 deaths annually [1,2]. The risk factors for renal cancer include smoking, alcohol intake, obesity, renal damage, and poor diet. Among various renal disorders, chronic kidney disease (CKD) is a fatal condition that affects between 5% and 7% of the world’s population and is a significant predictor of end-stage renal disease, cardiovascular morbidity, and ultimately death [3]. Regardless of the underlying cause, nearly all renal disorders result in permanent loss of kidney function due to gradual and irreversible nephron loss and diminished regeneration ability. The currently available chemotherapeutic agents only slow the progression of disease, necessitating the development of novel therapeutic drug approaches [4].

Tyrosine kinases are an important member of the enzyme family that catalyzes phosphorylation of tyrosine residues in protein by using ATP, which is a crucial step for normal cell development and homeostasis [5]. Potential anti-neoplastic therapeutic targets of tyrosine kinase that have been examined include epidermal growth factor receptor (EGF-R) and vascular endothelial growth factor receptor (VEGF-R). The epidermal growth factor receptor (EGFR) belongs to the tyrosine kinase family that is overexpressed in various cancers, such as breast cancer, prostate cancer and colorectal cancer, and plays an important role in cancer proliferation, growth and angiogenesis [6]. EGFR inhibitors have been developed in recent years that bind competitively to the active site of tyrosine kinase, thus blocking their uptake of ATP [7]. The gefitinib and erlotinib have been developed as EGFR inhibitors and many other drugs are undergoing clinical trials [8]. Another enzyme i.e., vascular endothelial growth factor receptors (VEGFRs) play an essential role by mediating angiogenesis, cell proliferation and blood vessel formation [9]. The VEGF receptors consist of three types, VEGFR1 (FLT1), VEGFR2 (KDR/FLK1) and VEGFR3 (FLT4). Among these, VEGFR2 is overexpressed in renal cancer, breast cancer, liver cancer and its activation by VEGFR signaling pathway is involved in tumor angiogenesis [10]. Recent advances in drug discovery have developed drugs that block VEGFR2 signaling pathways and thus inhibit angiogenesis [11], but these inhibitors because various adverse effects, such as nausea, skin rashes, and hypertension due to their non-selectivity [12]. The activation pathway of VEGFR and EGFR involves angiogenesis and cell proliferation [13], and is mentioned in Figure 1 below.

Among various heterocyclic compounds, oxadiazoles are five-membered ring structures having one oxygen and two nitrogen atoms. Many oxadiazole derivatives have been reported to have anticancer potential, and three potent FDA-approved oxadiazole drugs with reported anti-cancerous properties are given below in Figure 1 [14,15].

In this study, we used structure-based virtual screening techniques to investigate the anticancer potential of 14 oxadiazole derivatives which have been reported earlier for their anti-inflammatory activity [16]. The purpose of the current study was to test the inhibitory potential of 1, 3, 4 oxadiazole derivatives against VEGFR2 for the treatment of renal cancer. The chemical structures of 14 oxadiazole derivatives are given in Figure 2. The IUPAC names of all derivatives are given in the Appendix A. 

To measure the anti-cancer potential of our selected compounds (**7a–n**), a comprehensive multicomponent in silico study was designed. The study was comprised of density functional theory calculations and molecular docking studies, followed by molecular simulation studies. The pharmacokinetic profile of all the compounds was determined by calculating ADMET properties.

## 2. Materials and Methods

The whole procedure for the synthesis of selected derivatives, including the essential conditions, has been published in our earlier work [16]. Briefly, the study comprised of density functional theory studies with a focus on global reactivity descriptors i.e., hardness and softness of compounds, ionization potential, electron donating and accepting power. Molecular docking studies and molecular dynamic simulations studies were performed. The pharmacokinetic profile of all the compounds was then determined by ADMET software.

### 2.1. Density Functional Theory

The density functional theory is an effective method to obtain the optimized structures of compounds and for HOMO LUMO analysis. The Becke-3-Parameter-Lee-Yang-Parr (B3LYP) theory and SVP basis set was used for high accuracy in vibrational spectra both in gas and solvent phase calculations with the Gaussian 09 program [17]. The Gauss View 6 program [18] was used to obtain HOMO and LOMO structures, and to estimate the energy of frontier molecular orbitals.

### 2.2. Molecular Docking

#### 2.2.1. Selection of Protein Targets

In order to execute the virtual screening process crystalline structure of VEGFR 2 and EGFR protein was downloaded from the RCSB protein data bank with the PDB ID: 3VHE for VEGFR 2 and 2GS6 for EGFR [19,20]. 

#### 2.2.2. Softwares Required

The following softwares; MGLtools, binary files of Autodock4 and Autogrid4 [21], BIOVIA’s Discovery Studio Visualizer [22], SeeSAR [23], chemdraw ultra [24] and chemdraw 3D pro [25] were used for molecular docking.

#### 2.2.3. Preparation of Protein

The downloaded protein from RCSB was further processed for executing autodock studies. All the heteroatoms along with co-crystal ligand and solvent molecules were removed from the protein molecule in BIOVIA’s discovery studio visualizer. The pure protein structure obtained was then prepared for docking with the help of autodock inbuilt tools by adding necessary polar hydrogen and Kollman charges to each atom and was saved in pdbqt format [26]. 

#### 2.2.4. Preparation of Ligand and Molecular Docking

The structures of all the compounds were drawn in chemdraw ultra by using their IUPAC names and then the energy minimization for all the structures was carried out using chem 3D pro. The compound structures were saved in SDF format. These structures were then converted to autodock acceptable format i.e., pdbqt through openbabel GUI software. Structure based virtual screening was then performed using autodock4 software. The grid box dimensions were kept same as for co-crystal ligand to generate grid parameter file i.e., x: y: z = −28.96 Å, −4.140 Å, −14.515 Å for VEGFR 2 and grid box dimensions for EGFR was x: y: z =142.128 Å, 26.78 Å, 52.91 Å. For the generation of docking parameter file, Lamarckian genetic algorithm (LGA) and in house force field named Autodock4Zn were used. The no. of poses was set to 100 and population size was 300 to ensure maximum validity and reliability of the scoring function was estimated from extensive literature review [27]. The prepared ligand library was then docked into the active site of both proteins separately [21]. The docking of all the compounds was reconfirmed by SeeSAR Analysis [23].

#### Visualization

The BIOVIA discovery studio visualizer was used for the analysis of ligand and protein interactions. All the 2D and 3D conformations were generated by adding autodock result files and protein pdbqt files in BIOVIA’s Discovery Studio. The various bonding and non-bonding interactions of the ligand and active pocket were identified accordingly.

#### Validation

The docking protocol was validated on the basis of the RMSD value and by re-docking the co-crystal ligand into the active pocket of protein. Only poses with the RMSD value of docking and experimental value of ligand less than 2.0 Å were accepted [28].

#### 2.2.5. Molecular Dynamics Simulations

Desmond from Schrödinger LLC suite, developed by Bowers and his team [29], was used to simulate protein ligand complex for 100 nanoseconds. In the case of protein and ligand complexes, docking investigations were the first step in preparing them for molecular dynamics modeling. The binding status of a ligand into the active pocket of protein was calculated using molecular docking studies under static conditions. MD simulations calculate the motion of atoms over the period of time by applying Newton’s classical equation of motion into the molecule’s binding position in a protein’s active site [30]. By employing MD simulations, it was possible to anticipate the ligand binding status in extreme physiological conditions [31,32].

The Protein Preparation Wizard or Maestro software was used to preprocess the protein–ligand complex, which includes optimization and energy minimization of the protein–ligand complex structure. All the files were created with the help of the System Builder tool. Because of its simplicity, TIP3P was chosen as a solvent model with an orthorhombic box (Transferable Intermolecular Interaction Potential 3 Points). The OPLS-AA 2005 force field was utilized to execute the MD simulation [33]. Counter ions were added to the models to make them behave in a neutral manner i.e., 0.15 M sodium chloride (NaCl) was added to mimic physiological circumstances. For the whole simulation, the NPT ensemble with a temperature of 300 K and a pressure of 1 atm was employed. Prior to the simulation, the models were made more flexible. After every 100 ps, trajectories were saved for inspection, and the simulation’s stability was confirmed by calculating the root mean square deviation (RMSD) of the protein and ligand over time.

#### 2.2.6. Cell Viability Assay

In order to estimate the anti-cancer potential, all the derivatives were tested against a human cervical cancer cell line (HeLa) and human breast cancer cell line (MCF-7), by using the already reported method of Mosmann (1983) and Nik and Otto (1990) [34,35] as discussed in our previously published article [36]. The experiment was carried out in 96-well flat-bottom plates with 90 μL of medium and 10 × 10^4^ cells seeded into each well. 100 μL of test chemical solution was added to each well, and the plate was incubated for 24 h at 37 °C and 5% CO_2_. The positive and negative control wells were seeded with 10 μL of standard drug (cisplatin) and 100 μL of cells medium (without chemical). After that, each well was pipetted with 10 μL of MTT reagent and incubated for 4 h at 37 °C. Then, 100 μL of 10% sodium dodecyl sulphate solution was added and the mixture was maintained at room temperature for 30 min with intermittent shaking. Finally, the optical density was computed. All tests were done in triplicate, and the findings were given as percent growth inhibition values, as previously described [36].

#### 2.2.7. ADMET Properties

Predicting ADMET properties is a crucial step in the drug development process. The online web server ADMETlab 2.0 was used to calculate the physicochemical parameters, medicinal properties, absorption, distribution, metabolism and toxicity of derivatives. It is an accurate and reliable online platform for predicting ADMET properties [37].

## 3. Results and Discussion

### 3.1. Synthesis of 1, 3, 4-Oxadiazole Amide Derivatives

In our previously reported study [16], the target compounds **7a–n** were synthesized in a series of steps. First, 5-chlorophenyl-1, 3-dimethyl-4-oxadiazol-2-thiol (3) and 2-bromo-N-[aryl/aralkyl]propionamide (**5a–n**) were produced separately. The following steps were used to produce compound 3: The compound 4 chlorobenzoic acid (a) was first refluxed in ethanol under the catalysis of concentrated H_2_SO_4_ to produce the corresponding ethyl ester (1), which was then reacted with hydrated hydrazine in methanol to produce carbohydrazide (2), which was then refluxed with carbon disulfide in the presence of ethanolic potassium hydroxide to yield the cyclized product 5 (4 chlorobenzoic acid (3). The other aryl/aralkyl amine precursors (**5a–n**) of the target compounds were synthesized by combining the aryl/aralkyl amines (**4a–n**) with 2 bromopropionyl bromide in basic conditions (pH 9–10). The required compounds (**7a–n**) were successfully synthesized in high yield by treating the precursor 3 with the electrophiles 2-bromo N [aryl/aralkyl] propionamide (**5a–n**). Infrared, nuclear magnetic resonance (NMR), and 13C NMR spectroscopy, as well as electron ionization mass spectrometry (EIMS) and high-resolution electron ionization mass spectrometry (HREIMS) [16] were used to determine the structures of substances (Figure 3).

### 3.2. Density Functional Theory Calculations (DFTs)

All the DFT Calculations were carried using Gaussian-09 via DFT35 with B3LYP36 Hybrid GGA functional theory and SVP basis set. The Table 1 shows optimized geometry parameters in gaseous and ethanol phase.

#### Results of DFT Studies: Global and Local Descriptors

The table of global reactivity descriptors reveals that, according to the ionization potential data, **7j** and **7n** require less energy to remove an electron from the ground state, whereas **7b** required the highest energy indicating the highly reactive nature of **7j**. However, the value of this attribute for **7j** and **7b** was 0.2300 and 0.24068 eV, as estimated with SVP basis set. The compound **7j** has the highest value of LUMO and HOMO energy gap representing the highest stability of this derivative. The electronegativity value indicated that **7b** has the maximum ability to attract electrons in both phases due to negligible effect of solvent. This compound would be energetically favorable for a nucleophilic attack. The findings of all parameters are significantly different for each of the compounds, as shown in Table 2.

The electron-accepting and electron-donating power of the various oxadiazoles as well as their tendency to donate or accept a small amount of charge required for chemical interactions, were determined. Chemical hardness of any compound is an important parameter to determine its reactivity potential and compounds with higher values of hardness are less prone to be reactive. Among 14 oxadiazoles, most of the derivatives exhibited almost similar hardness values and are within the acceptable range of reactivity. The values of all other parameters for all the compounds are given below in Table 3.

The optimized structure of top 3 potent compounds having Frontier molecular orbitals (HOMO & LUMO) along with energy gap is presented in Figure 4, while the other compounds are given in Appendix A.

### 3.3. Molecular Docking

The crystalline structure of VEGFR 2 indicated that it is separated into two lobes by its folded form. The phosphotransfer catalysis takes place in an inter lobar gap between the two lobes. When the N-terminal lobe (residues 820–920) is folded, one helix and one twisted beta sheet structures are formed (aC). The b structure is made up of five antiparallel strands (b1–b5), three of which are severely coiled and curl over the other two strands (b4–b5), while the other two strands are straight. The larger C-terminal domain (residues 921–1168) has two antiparallel b strands (b7–b8) near the N-terminal b sheet at the top of the C-terminal domain. At the bottom of the C-terminal domain is the N-terminal b sheet. The C-terminal domain’s basic structure is made up of seven a helices (aD, aE, aE-F, aF, aG, aH, and aI). The glycine-rich nucleotide binding loop (residues 841–846), the catalytic loop (residues 1026–1033), and the activation loop (residues 1046–1075) are all functionally essential loops for the proper functioning of VEGFR 2.

The active pocket of VEGFR 2 consists of the following sequence of amino acids i.e., Asp814, Cys817, Leu840, Val848, Ala866, Lys868, Ala881, Leu882, Ser884, Glu885, Ile888, Leu889, Ile892, Val898, Val899, Val914, Val916, Glu917, Phe918, Cys919, Lys920, Phe921, Gly922, Asn923, Leu1019, Cys1024, Ile1025, His1026, Arg1027, Leu1035, Ile1044, Cys1045, Asp1046, Phe1047, Gly1048, and Leu1049 [38] and the active pocket of EGFR protein consists of the following sequence of amino acid Leu694, Gly695, Ser696, Phe699, Val702, Ala719, Ile720, Lys721, Leu723, Lys730, Ala731, Lys733, Glu734, Ile735, Asp737, Glu738, Val741, Met742, Cys751, Leu764, Ile765, Thr766, Arg812, Asp813, Arg817, Asn818, Leu820, Thr830, Asp831, Phe832, Gly833, Leu834, Ala835, Lys836, Tyr845, Ala847, Glu848, Gly849, Gly850, Ys851, Val852, Pro853, Arg865, and Tyr867. The Tyr6 is only amino acid from 2^nd^ chain that is involved in active pocket activation [39].

#### 3.3.1. Structure Activity Relationship (SAR) of 1, 3, 4-Oxadiazole Amide Derivatives

The binding energies of all the derivatives were determined against VEGFR2 and EGFR. All the derivatives showed strong interactions within the active pocket of VEGFR2 and exhibited less binding score against EGFR. The differential analysis of molecular docking was based on the fact that any potent derivative with strong interactions has minimum binding energy [26]. The structure activity relationship was studied on the basis of binding interactions, binding scores and predicted inhibitory concentration values obtained during docking analysis (Table 4). 

Among all the derivatives, the parent compound i.e., **7a** exhibited high binding energy value and less estimated inhibitory constant value among all the derivatives. The compound **7j** (−48.89 kJ/mol) which was found as the best inhibitor has methyl substitution at 3 and 5 position of the benzene ring. When the structure and activity of compound **7j** was compared with other derivatives it was found that the substitution of methyl group at meta position is responsible for the inhibitory potential of this compound. This effect can be seen in case of compound **7g** (−46.32 kJ/mol) which was found as the second most potent inhibitor of VEGFR2. In this compound the substitution of methyl was done at ortho and meta position i.e., 2nd and 5th position of phenyl ring. It can be suggested that the shifting of one methyl from 3rd to 2nd position resulted in the slight difference in inhibitory potential of compounds. Both compounds **7j** and **7g** exhibited equipotent Estimated inhibitory concentration values i.e., 0.0009 μM. The methyl group is electron donating group which imparts a positive mesomeric effect (*+M*) reasonable for inhibitory potential. The effect of substitution was compared between those derivatives in which ortho substitution was carried out i.e., derivative **7b**, **7d**, **7e** and **7f**. The compounds **7b** and **7d** have methyl substitution at ortho position which was found less favorable for the ligand-protein interaction as well as Estimated inhibitory values were found less potent. On the other hand, compound **7e** and **7f** exhibited better results. The detailed analysis suggested that the compound having substitution at ortho position alone (either at position 2 or position 4) or di substituted (2 and 4 position) were found less potent whereas, the compound with di-substitution at one ortho and one meta showed better result suggesting that in these derivatives. the substitution with meta group resulted in improved binding interaction as well as Estimated inhibitory constant value. An interesting behavior was observed when the substitution at ortho position was done with the bulkier group i.e., ethyl and ethoxy as in case of **7k** and **7l** (ethyl substitution); **7m** and **7n** (ethoxy substitution). All these compounds were found less potent as compared to derivatives with less electronegative CH_3_ group. 

The structure activity relationship of these derivatives was also studied against EGFR, and it was observed that these compounds showed less potent behavior against EGFR with higher binding energy and less inhibitory potential as depicted by their estimated inhibitory concentration values. However, the pattern of the substitution effect was same as observed in VEGFR2. The compounds having meta substitution were found more potent than para and other substitution. 

#### 3.3.2. Binding Interaction Studies

##### Binding Interactions of 1, 3, 4 Oxadiazoles with VEGFR2

In order to study the binding interactions of these derivatives, first of all the co-crystal ligand was re-docked within the active pocket of the selected protein (Figure 5). Here, compound 1-{2-fluoro-4-[(5-methyl-5H-pyrrolo [3, 2-d] pyrimidin-4-yl) oxy] phenyl}-3-[3-(trifluoromethyl) phenyl] urea was the co-crystal ligand for our target protein. The re-docking analysis suggested that this compound formed 3 covalent bonds by the three fluorine atom attached to the methyl phenyl ring with three amino acid residues i.e., His1024, Ile1044 and Cys1045, respectively. This ring was also involved in forming one sigma bond with Leu889 and another sigma bond was formed by 6 membered pyrimidine rings with Leu1035 amino acid residue. This derivative formed 5 conventional hydrogen bonds with different amino acid residues. The two amine groups of phenyl urea ring formed two hydrogen bonds with the same amino acid residue, Glu885, the carbonyl oxygen formed one hydrogen bond with Asp1046, the nitrogen atom of pyrimidine ring formed one hydrogen bond with Cys919 and one hydrogen bond was formed between fluorine atom and His1026 amino acid residue. The methyl group attached to the pyrole ring was involved in making one carbon hydrogen bond with Leu840. This Leu840 also formed one sigma bond with pyrole ring and one weak alkyl bond and one pi-sigma bond with pyrimidine ring. Besides these bonds, single pi-pi T shaped bond (phe1047) was formed with benzene ring present adjacent to pyrimidine ring. The other with various weak alkyl and pi-alkyl bonds were formed with Val898, Ile1044, His1026, Lys868, Val916 and Val848.

The most potent derivative i.e., **7j** formed different types of bonding and non-bonding interactions with Phe1047, Val916, Ile1044, Val848, Leu1035, Ala866, Phe918 and Leu840. This derivative formed 2 pi-sigma bonds, 2 hydrogen bonds, 3 pi-alkyl interactions and one pi-anion interactions. Regarding 2 sigma bonds, one sigma bond was formed between xylene ring and Leu889 and one sigma bond was formed between 1-chloro-4-methylbenzene ring and Leu1035. This 1-chloro-4-methylbenzene was also involved in making three pi-alkyl interactions with Ala866, Cys919 and Phe918 amino acid residues. Among two hydrogen bonds, one was formed between N-methylformamide group and Glu885, and second hydrogen bond was formed by the oxygen with Asp1046. Moreover, among 3 pi-alkyl interactions, one pi-alkyl interaction was formed by oxadiazole ring with Val848. This oxadiazole ring was also involved in forming one pi-anion interaction with Cys1045. The amino acids involved in non-bonding interactions were: Val916, Val848 and Leu840 amino acid residues.

The second most potent derivative i.e., **7g** formed different types of bonding and non-bonding interactions with Leu1035, Leu889, Cys1045, Glu885, Val899, Lys868, Asp1046, Ile892, Phe1047, Val916, Ile1044, Val848, Leu1035, Ala866, Phe918 and Leu840. This derivative formed 3 pi-sigma bonds, 2 hydrogen bonds, 8 pi-alkyl interactions, one pi-anion and one pi-cation interactions. The chlorobenzene ring formed one pi-sigma interaction with Leu889 and one pi-alkyl interaction with Ile892. The oxadiazole ring was involved in forming Hydrogen bond interaction with ASP1046, pi-anion interaction with Glu885, Lys868 and pi-alkyl interaction with Val899. Moreover, N-propylacetamide formed strong hydrogen bond with Lys721. The toluene ring showed three pi-alkyl interactions Ala866, Cys919 and Val848 with amino acid residues. The toluene ring was also involved in forming one pi-sigma interaction with Leu1035. 

The ligand interaction of compound **7l** with the active pocket rediues showed that the chlorobenzene ring formed pi-alkyl interactions with Ala866, Leu840, Cys919, Phe918 amino acids and also formed one pi-sigma interaction with Leu1035. Moreover, the oxadiazole ring was involved in forming pi-anion interaction with Cys1045 and one Pi-alkyl interaction with Val916. The toluene ring formed one pi-sigma interaction with Leu889 and two pi-alkyl interaction with His1026 and Leu1019 amino acid residues. The 3D and 2D binding modes of most potent compounds **7j**, **7g** and **7l** are mentioned in Figure 6 and Figure 7 while 2D, 3D interactions of other compounds are given in Appendix A.

##### Binding Interactions of 1, 3, 4 Oxadiazoles with EGFR

Following the same method as with VEGFR protein, the co-crystal ligand i.e., analogue of ATP (Thiophosphoric acid o-((adenosyl-phospho) phosphor)-s-acetamidyl-diester) of EGFR protein was re-docked within the active pocket (Figure 7). The various amino acid residues which were involved in bonding and non-bonding interactions are: Ala698, Phe699, Asp831, Asn818, Arg817, Glu738, Phe832, Met769 and Lys721. The analysis of docking results suggested that 11 conventional hydrogen bonds were formed between co-crystal ligand and different amino acid residues of active pocket. The two-hydroxyl group of pentose sugar formed four hydrogen bonds with Asp831, Asn818, Arg817 amino acid residues, one hydrogen bond was observed between oxygen atom and Lys721. One conventional hydrogen bond and one carbon hydrogen bond was formed between Ala698 and nitrogen atom of adenine ring. In the same way, Phosphate group adjacent to adenosine moiety formed two hydrogen bonds with Glu738 and Phe832, respectively. Moreover, acetamides group formed two hydrogen bonds with Met769 amino acid residue. The co-crystal ligand showed one pi-sigma bond by adenine ring with Phe799 and also exhibited two cationic bonds by electropositive Phosphorus atom with Asp831 and Glu738, amino acid residues. 

According to the docking results **7j** is the most potent compound of the series. The compound **7j** formed different bonding and non-bonding interactions with following amino acids i.e., Ala719, Leu768, Met769, Val702, Leu820, Thr766, Lys721, Leu764, Glu738, Thr830 and Asp831. The most common bonding interactions included hydrogen bonding, pi-sigma bonds, weak cationic bonds and pi-alkyl bonds. The docking results of **7j** suggested the presence of 3 hydrogen bonds, 1 pi-sigma bond, 2 cation bonds and 5 pi-alkyl bonds. The amino acid residues Thr766, Lys721 and Thr830 formed 3 hydrogen bonds with nitrogen atom of propanamide group, oxygen atom of oxadiazole ring and 1st nitrogen atom of oxadiazole ring. The pi-sigma bond was formed by the xylene ring and Leu820. The two cationic bonds were formed by the Glu738 and Asp831 with the oxadiazole ring and adjacent phenyl ring. In addition to these 5 weak pi-alkyl bonds were also formed by xylene ring and 3 amino acid residues i.e., Ala719, Leu768, and Met769.

The docking interaction of 2nd best compound **7g** involved the following amino acids residues i.e., Thr766, Leu764, Lys721, Ala719, Leu694, Leu768, Gly772, Met769, Leu820, Thr830 and Asp831. The analysis of docking results revealed the presence of one conventional hydrogen bond, one carbon hydrogen bond, two pi-sigma bonds and seven pi-alkyl bonds. The carbon hydrogen bond was present between Thr766 and xylene ring while conventional hydrogen bond was formed by the Lys721 and oxygen atom of amide group. One pi-sigma bond was formed by amino acid Leu820 and oxadiazole ring and other pi-sigma bond Leu694 and phenyl ring. The weak pi-alkyl bonds were formed by 5 amino acid residues i.e., Leu764, Lys721, Ala719, Leu768 and Leu820.

The sequence of amino acid residues involved in the docking interaction of **7l** are Asp831, Thr830, Leu820, Met769, Leu694, Ala719, Leu768, Gln767, Met742, Lys721 and Thr766. The bonding and non-bonding interactions of **7l** included 2 conventional hydrogen bonds, 1 carbon hydrogen bond, 1 pi sigma bond, 4 pi-alkyl bonds and 1 sulphur bond. The only carbon hydrogen bond was formed by the Asp831 and ethyl benzene ring. In the same way one conventional hydrogen bond was formed by Thr830 and Lys721 with the hydrogen and oxygen atoms of propanamide group. The only pi-sigma bond was formed by Leu820 and Phenyl ring and the sulphur bond was formed by the sulphur atom of Met742 and oxadiazole ring. The pi-alkyl bonds were formed by Leu768, Leu694, Ala719 and Leu820 with the ethylbenzene ring and methyl group attached to phenyl ring. The 3D interactions of the most potent derivatives are shown in Figure 6 while 2D and 3D interactions of other derivatives are given in the Appendix A.

### 3.4. SeeSAR Analysis

The SeeSAR study was used to compute the HYDE (Hydrogen bonding and Dehydration) score using BiosolveIT’s SeeSAR [23]. The computation of desolvation value, hydrophobic interaction, and hydrogen bonds established between the ligand and active pocket are all part of the HYDE analysis of a molecule. HYDE also aids in anticipating the specific region of the complex that experiences favorable and unfavorable ligand receptor binding and is represented by green and red coronas around the specific atom. The larger coronas gave higher HYDE value representing greater involvement in binding affinity. All the atoms (dark green sphere) that contributed favorably towards binding affinity and their individual HYDE values for the best molecule are highlighted for the most potent compounds. The role of atoms in each ligand must be determined to anticipate overall binding affinity or interactions with other molecules. It is evident from the results that specific region of each ligand contained few atoms that contribute positively toward the binding affinity. The HYDE value of all potent compounds is greater for VEGFR 2 than EGFR (Figure 8).

### 3.5. Molecular Dynamics Simulations

The protein-ligand complexes of VEGFR and EGFR with the most favorable docking pose of **7j** were simulated in aqueous conditions for 100 ns using Desmond by Schrodinger. The structural behavior of both proteins and their complexes was determined by Root Mean Square Deviation (RMSD), Root Mean Square Fluctuation (RMSF), Solvent accessible surface area (SASA) and Radius of gyration (Rg) along with other MD simulation parameters. The RMSD graph provides the detail of any structural fluctuation occurred because of ligand-protein interaction. For the analysis of MD simulation results the RMSD plot of both proteins and their complexes was generated and compared to evaluate the stability of protein complex. Figure 8 shows the RMSD graphs of both proteins along with their complexes. The RMSD graph of VEGFR complex showed some initial fluctuation and then became stable after 30 ns at RMSD value of 1.8 while the graph of EGFR showed that the protein is relatively unstable, and its **7j**-complex became stable after 50 ns with RMSD value of 2.5. The data clearly shows that VEGFR 2 and its complex was found more stable in aqueous medium as compared to EGFR which augment the docking results (Figure 9). 

Further analysis of MD simulation was carried out by generating RMSF plots, and the results are presented in Figure 10. Any fluctuation in the amino acid residues of C and N terminal lobe in target protein can be identified by RMSF graphs. The average RMSF value for C_α_ chain with majority of amino acid residues of VEGFR was found to be less than 1 Å and for the residues of EGFR protein was below 1.2 Å suggesting higher stability of VEGFR protein. The very few residues of the protein VEGFR, on the other hand, experienced small variations, which could be related to the hanging position. The RMSF data further established the stability of VEGFR proteins and its complex in aqueous conditions. The RMSF value of both ligands exhibited some fluctuations which showed their dynamical shift at their binding site in respective proteins. (Further 2 replicates of 50 ns for each protein has been added in the Appendix A).

The radius of gyration (Rg), solvent accessible surface area (SASA), MSD value and intra-molecular hydrogen bonding of both systems was calculated by analyzing MD simulation data. The mass-weighted RMS distance of atomic cluster from their common center of mass is known as Radius of gyration (Rg). In MD simulation studies Rg is an important factor in finding the compactness of protein [40]. Rg value of VEGFR 2 was found to be uniform throughout the simulation length, demonstrating their stability with an average value of 5.2 and for EGFR it was 4.0. The Rg of both proteins remained constant during the simulation, indicating that the systems did not experience any notable conformational changes [41]. SASA is another factor used in MD simulations to assess protein stability [42]. Throughout the simulation period, SASA of VEGFR 2 and EGFR were remained stable. The average SASA of VEGFR 2 and EGFR was 55Å and 160Å, respectively. Figure 11 summarizes different graphs for both proteins.

Moreover, the alpha-helices and beta-strands were considered as secondary structure elements during the simulation (SSE). The distribution of SSE across the protein structure by residue index is given in Figure 12. The figure below shows the SSE composition for each trajectory frame during the simulation, SSE assignment for each residue over time. The residue index of VEGFR ranges from 60–70, 140–155 and 230 to 300 belongs to beta strand. The % SSE graph of VEGFR complex shows the least fluctuation and is comparatively more stable than EGFR. 

At the end, the protein interactions with the ligand were also observed during the simulation process. These developed linkages were identified by four types of protein-ligand interactions i.e., hydrogen bonds, hydrophobic interactions, ionic interactions, and water bridges. The ‘Simulation Interactions Diagram’ panel in Maestro was used to study the subtypes of each interaction type. Over the course of the trajectory, the stacked bar charts were standardized: for example, a value of 0.7 indicated that the specific interaction has been maintained for 70% of the simulation duration. Due to several interactions with the same subtype of ligand, values above 1.0 were observed which is also feasible.

Interestingly, the hydrogen bonds observed during the study exhibited a considerable influence on drug selectivity, metabolism, adsorption, and hydrogen-bonding properties that is important in drug designing [43,44]. Here, the four types of hydrogen bonds included: backbone acceptors, backbone donors, side-chain acceptors, and side-chain donors that exists between a protein and a ligand (Figure 13).

The majority of significant ligand–protein interactions detected by MD simulations are water bridges and hydrogen bonds, as shown in Figure 12. In terms of hydrogen bonding, ASP-1046 and CYS-919 are the most significant residues for 7j-VEGFR 2, while MET-769 is the most critical amino acid for 7j-EGFR complex. Over the course of the trajectory, the stacked bar charts were standardized: for example, a value of 1.0 indicates that the specific interaction was maintained for 100% of the simulation time. Many protein residues may have several interactions with the ligand of the same subtype, therefore values above 1.0 are also attainable. Figure 14 depicts individual ligand atom interactions with both protein residues. Only those interactions that occur in the chosen trajectory (100.0 ns) for more than 30.0 percent of the simulation duration are mentioned.

### 3.6. Cell Viability Assay

To support the in silico studies, the preliminary screening of the most potent derivatives was carried out using in vitro cell viability assay i.e., MTT assay (Table 5). 

The human cervical cancer cells (HeLa) and MCF-7 cells were treated with 100 µM concentration for 24 h and 48 h. The concentrations were selected based on the predicted inhibitory values obtained during docking studies (as shown in correlation plot Figure 15). 

A time dependent linear response of cell death was observed at single dose concentration. The compounds showed good results and caused % growth reduction but the derivatives **7c**, **7g**, **7h**, **7j**, **7l**, **7k** and **7m** showed maximum cell death justifying the computational studies where these compounds were found to be the best inhibitors of VEGFR-2 and also EGFR. The single concentration cisplatin was used as positive control. The results were calculated by comparing with the total activity control (without inhibitor) i.e., un-treated cells. The % viability graph was generated via graph pad Prism software and is given in the Figure 16.

### 3.7. ADMET Properties

The ADMET (absorption, distribution, metabolism, excretion, and toxicity) properties of all the VEGFR 2 inhibitors were calculated to compute its appropriateness as a drug molecule. Other physicochemical properties explored are mentioned in our previous paper [45].

The physiochemical attributes of a clinically active substance have a substantial impact on its ability to become a therapeutically effective medicine. Compounds with MW < 500, nON < 10, nNH < 5, nRot < 10, and cLogP < 5 are considered orally bio-available and have a good ADMET profile [37]. The most powerful molecule, **7j**, has an excellent physicochemical profile, according to the findings. The physicochemical properties of compounds are mentioned in Table 6.

According to ADMET results of absorption and distribution, the compound **7k** has maximum volume of distribution. Other properties like blood–brain barrier, placenta barrier, human intestinal absorption, plasma protein binding, P-gp inhibitor and P-gp substrate indicated that **7a**, **7j** and **7g** are pharmacokinetically the most suitable. The values of other compounds for all these properties are given in Table 7.

The metabolism and excretion profile of all the 1,3,4 oxadiazoles suggested that most of the potent derivatives are the best target for liver enzymes i.e., CYP1A2 and CYP2C19 while all other liver enzymes are best complexed with different compounds. The value of **7m**, **7c** and **7j** is highest for CYP2C9, CYP2D6 and CYP3A4. The ADMET results suggested that the compound **7h** is excreted slowly from the body and the small half-life of **7i** is responsible for its shortest stay in the body (Table 8).

The medicinal properties of a compound explained the potential of a candidate molecule to become a drug. The data showed that all the 14 compounds are in agreement with Lipinski rule of 5 which included molecular weight, donors and acceptors of H bonds, and partitioning coefficients (log P). The synthetic accessibility score of all compounds was found below 3.0, indicating the ease of synthesizing them (Table 9).

The toxicity profile of any molecule is a paramount criterion for its selection regarding further studies. The drugs with high toxicity cannot be considered for human trials which is indicated by −sign while +sign indicates acceptable results. The results of our study showed that **7e** and **7i** are most toxic and **7a** and **7n** are least toxic among the given oxadiazoles derivatives while all the compounds are equally dangerous for eyes. The data showed that **7g**, **7j** and **7n** are least toxic. All the values are given in Table 10.

As a result, we can conclude that our molecules, particularly the most potent derivatives **7g**, **7j** and **7l** exhibit drug-like properties. Further research is required to investigate toxicity profiles of given compounds. 

## 4. Conclusions

The findings of current study convincingly suggested that selected 1,3,4 oxadiazole derivatives are potent inhibitors of VEGFR 2 compared to EGFR and can be used in the treatment of cancers where selective VEGFR2 inhibitors are required i.e., renal cancer and to mitigate angiogenesis in other cancers. According to the results of the molecular docking investigations, density functional theory, cell viability assays and MD simulation, **7g**, **7j** and **7l** are the most potent compounds and should be further explored for the development of new molecules.

## Data Availability

The required data will be available upon reasonable request. All the relevant is included in the Appendix A.

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
