# Peer review of "Computational Investigation of 1, 3, 4 Oxadiazole Derivatives as Lead Inhibitors of VEGFR 2 in Comparison with EGFR: Density Functional Theory, Molecular Docking and Molecular Dynamics Simulation Studies"

_biomolecules, 2022, doi:10.3390/biom12111612_

Round 1

Reviewer 1 Report (New Reviewer)

The authors reported a computational study of various potent VEGFR inhibitors. The manuscript's content is acceptable for publishing in biomolecules. I have a few comments for the authors.

1. The caption of Scheme 1 should be expanded. More information should be included. For instance, what are Flt1, ERT, SOS, etc...? What do those double vertical lines mean? Do thin arrow and thick arrow mean differently? 

2. In the section 2.3.2 Molecular Dynamics simulations, it states that the simulation is performed for 100 nanoseconds. How do the authors know whether 100ns is long enough? What is commonly used in the field?

3. Also in the section 2.3.2, the sentence "Prior to the simulation, the models were made more flexible". What do authors mean by "flexible"?

4. Table 4 is not referred in the text

5. In general, Figure captions need to be expanded with more descriptions

6. Line 510-511. What does SSE mean?

7. Doe the bars in Figure 7 indicate the percentage? If so, why is there a bar that exceeds one?

Author Response

Reviewer 2 Report (New Reviewer)

In this manuscript, the authors applied both QM and MM methods to study drug properties of 14 chemicals as potential VEGFR2 inhibitors. The authors are suggested to take the following comments into consideration to refine their work.

1. Authors need to carefully check their manuscript for typos and misspellings. E.g, in the title, "Molecular Dynamic" should be "Molecular Dynamics"; "and 24068 eV" on line 241 should be "and 0.24068 eV".

2. Double check the LUMO and HOMO data for 7i and 7c. It's weird that the trend get reversed for these two compounds compared to others. I think their data get misassigned.

3. For QM, did the author check IR frequencies and make sure the conformation they got is really minimum (without imaginary frequencies) for each chemical?

4. The authors need to carefully make their SAR conclusions. E.g. considering" "When the structure activity relationship of compound 7j was compared with other derivatives it was found that the substitution of methyl group at meta position is responsible for the maximum inhibitory potential of this compound".  However, considering compounds 7b, 7c, 7d which has methyl group single substitution,  for HeLa cell line result,  activity is in order of 7d>7b>7c (para>ortho>meta), and for MCF-7, 7d>7c>7b (para>meta>ortho).  For double substituted chemical 7h and 7j, double meta seems to be more potent than double ortho substitutions. So there is some kind of ambiguity in the data to establish a clear SAR. However, all data support that methyl substitution is good for potency when comparing data of 7a which has no substituents with others.

5. The authors can make a correlation plot between docking score and experimental potency. The docking score seems to work pretty well here. It predict 7j, 7g and 7l are the most favorable binders while the experimental data supports this.

Round 2

Reviewer 2 Report (New Reviewer)

The authors addressed all my comments, however, table 2 need to be revised. After fix the homo and lumo value for compound 7c and 7i, ω- and ω+ as well as Δω± need to be re-evaluated, and for 7i, the authors forgot to recalculate ∆Egap. It can be published after the authors fix this table without further review.

Author Response

This manuscript is a resubmission of an earlier submission. The following is a list of the peer review reports and author responses from that submission.

Round 1

Reviewer 1 Report

The manuscript of Sajjad Bila et al. presents a computational study on a small library of previously reported 1, 3, 4 oxadiazole derivatives. 
From a technical point of view all the calculations appear to be correctly performed, even if not adopting advanced methodologies. 
However, the work suffers a couple of very relevant problems.  

First, the use of docking to demonstrate the activity of a series of compound to a target can not be acceptable. Molecualr docking is a powerful tool to prioritize compounds in a virtual screening or to characterize the binding mode of a well-validated ligand. On the contrary,  conclusion of the authors seems to convince the reader that we have a conving series of inhibitors based on a very raw scoring function.

There is a plethora of publications on the weakness of scoring function, and more advanced methods were proposed to have an idea about the strength of the interactions.  For instance, Relative binding free energy (RBFE) calculations are methods that provide more reliable predictions.
 If the author would convince that their protocol is sufficiently reliable in this task they should at least provide a benchmark of already known inhibitors that are correctly ranked( or at least classified)  among a series of not binders. ChEMBL and BindingDB offer  huge dataset of experimental data to validate the protocol.
Other critical point, since all we know about the inaccuracy of scoring function...which is the meaning in using double digit decimals ?
For instance, the two reported scoring are not in agreement in table 4. ...and how was the pIC50 predicted?
other some minor points

the  figure 16  was already reported in Ref. 16
there is a figure on page 3 without a legend. 

In my opinion, the paper cannot be accepted without any solid validation.

Reviewer 2 Report

The works focus on the discovery of novel selective inhibitors against VEGFR, an important target for renal cancer

The biggest limitation of this work is that it is purely computational. As a computational biochemistry myself, I find that the computational techniques are very powerful tools to explain, to the atomistic level, in vitro experiments. By itself, I would say that molecular docking techniques have a high chance to fail and the conclusions of the work are gambling. 

The energies of the docking calculations are orientative and should not be translated in inhibition constants. I would suggest the authors to test at least the best compounds in vitro in order to validate the conclusions they claim to achieve. I was very surprised with the solvent exposure of hydrophobic substituents (see Fig. 13). Regarding, the Molecular dynamics simulations, I would like to see the results of more replicates of each complex (at least 3 runs per complex). 

Moreover, the manuscript figures should be improved.